# Effect of Double Pulse Resistance Spot Welding Process on 15B22 Hot Stamped Boron Steel

**Hwa-Teng Lee \* and Yuan-Chih Chang**

Department of Mechanical Engineering, National Cheng Kung University, Tainan 701, Taiwan;
k3341688@gmail.com
\* Correspondence: htlee@mail.ncku.edu.tw; Tel.: +886-6-2757575 (ext. 62154)

**Abstract:** Double pulse resistance spot welding process by applying a second step welding current is a new pathway to alter the mechanical properties for advanced high strength steels. Herein, the resistance spot welding (RSW) of hot stamped boron steel 15B22 by one-step and two-step welding with different welding currents is investigated. The results of the tensile–shear test, size of the weld nugget, hardness distribution, microstructure, and failure mode of different welding parameters are analyzed. The weldment of the two-step RSW with a higher heat input exhibits a lower tensile–shear load and lower fracture energy when the size of the weld nugget is large. The microstructural study reveals the appearance of a partially melted zone and sub-critical heat affected zone in the weldment where the fracture readily occurred. Thus, the two-step RSW process weakens the strength of the sample, which is attributed to the partial softening in the weldment due to the higher heat input.

**Keywords:** double pulse resistance spot welding; hot stamped boron steel; heat affected zone softening; failure mode

---

## 1. Introduction

In order to reduce carbon emissions and energy consumption, the use of advanced high strength steels (AHSSs) is being employed in the automotive industry. These steel grades demonstrate higher mechanical properties than conventional high strength low alloys (HSLA) steels and could fulfil the requirements of the different parts of the automotive body. The 15B22 is a hot stamped steel that utilizes boron additives to increase the hardenability and is commonly used for safety parts, such as A- and B-pillars [1]. After the hot stamping process, the matrix of the steel could transform into a fully martensitic structure and results in tensile strengths of up to 1500 MPa. When performing the joining process, the RSW is the most common method because of its low cost and stability. However, the difference in properties, such as the electrical resistivity or the content of the martensite, causes the spot welding performance of AHSS to be different from that of HSLA [2]. Previous studies have indicated that the AHSS has a high tendency to undergo expulsion, thereby resulting in a decrease in the tensile–shear load and energy absorption [3,4]. Furthermore, the expulsion would form a burr, which causes difficulty and increases the cost of the subsequent processes. Therefore, expulsion is not acceptable in the automotive spot-welding process. In addition, the AHSS has a narrower lobe diagram that results in fewer optional welding parameters [5], and the interfacial failure (IF) of AHSS tends to be higher than that of HSLA [6]. The use of IF as a quality criterion is still an argument. For AHSS, it may be able to meet the tensile shear force requirement of the spot welding standard even in the IF failure. But even so, the appearance of the IF failure means the lower heat input and lower mechanical properties, which should be avoided as much as possible [3].

In recent years, a new multi-step RSW process that differs from the conventional one-step RSW has been developed. The process uses a pulsed welding current that causes the weldment to undergo

multiple heating and cooling effects to produce a different thermal history. Hwang et al. [7] highlighted that the multi-step RSW could increase the range of the lobe diagram and reduce the expulsion; in addition, the change in the thermal history could impact the failure mode. Jahandideh et al. [8] demonstrated that the multi-step RSW would reduce the IF tendency of the HSLA. Nevertheless, the effect of the multi-step RSW process on the 15B22 remains unclear. On the other hand, it has been demonstrated that the spot welds that exhibit heat affected zone (HAZ) softening during welding, such as DP780 and DP980, tend to fail in this softened zone and change the failure mode [9,10]. Also, Mohamadizadeh et al. [11] reported that there were two significant softening regions in the HAZ and the fusion boundary. Sherepenko et al. [12,13] and Li et al. [14] additionally studied the issue of the softening of the fusion boundary. Xia et al. [15] concluded that the higher the volume fraction of the martensite, the greater the degree of softening. It is necessary to consider the effects of softening on the hot stamped steel with a fully martensitic structure. The main objective of this study is thus to investigate the softening effects of the hot stamped boron steel 15B22 by one-step and two-step RSW welding procedures and their impact to the mechanical properties. The mechanical properties and microstructure after welding under different conditions were further examined and discussed.

## 2. Materials and Methods

The material used in this study is uncoated 15B22 hot stamped boron steel (1.2 mm thick). The sheet was first heated up to 930 °C in the furnace for 5 min, then was transferred to the die for quenching. The die holding time was 15 seconds, and the temperature of cooling water for the hot stamping die was 27 °C. Shot peening was finally conducted for removing the oxide. The chemical composition is listed in Table 1. In order to confirm the welding range of the material, a pre-experiment was conducted to establish the welding current range, as shown in Figure 1. According to the AWS D8.1M automotive spot welding standard, the left limit of the welding current range is $4t^{0.5}$, where $t$ is the sheet thickness in millimeters. When the current is too low, the weld nugget is negligible and the property of the weldment is inadequate. When the current is too high, a liquid metal spray—termed an expulsion—is produced during the welding process. This state would produce a burr and has a negative effect on the process; hence, the temperature is set to the appropriate limit. In this study, a fixed weld time of 250 ms was used, with a welding range of 4.6 to 6.2 kA. The three main current values of 5.0, 5.5, and 6.0 kA from low heat input to high heat input are selected as the welding parameters of this experiment.

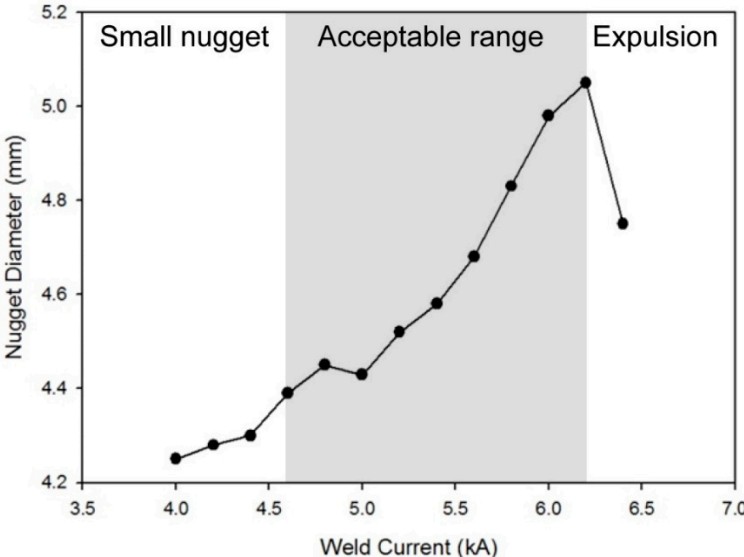

**Figure 1.** Welding current range of 15B22.

**Table 1.** Chemical compositions of 15B22.

| Element | C | Si | Mn | P | S | Al | B | Fe |
|---------|------|------|------|--------|--------|--------|--------|------|
| wt% | 0.22 | 0.20 | 1.20 | <0.020 | <0.010 | <0.075 | 0.0017 | Bal. |

As shown in Figure 2, the welding processes employed were the one-step and two-step RSW. The electrode force applied for both processes was fixed at 400 kgf, whereas the squeeze, weld, and holding times were 1000, 250, and 200 ms, respectively. Additionally, an additional preheating current (6.0 kA) was applied to the two-step RSW for 15 ms. The detailed parameter settings are shown in Table 2. The RSW welding tests were conducted using an intermediate frequency DC welding machine (NASTOA CO.,LTD., Tokyo, Japan) with Cu-Cr-Zr alloy electrodes. The electrode tip was customized, had truncated cone tips with a 6 mm diameter, and was cooled by a continuous circulating flow of water. The geometry was shown in Figure 3.

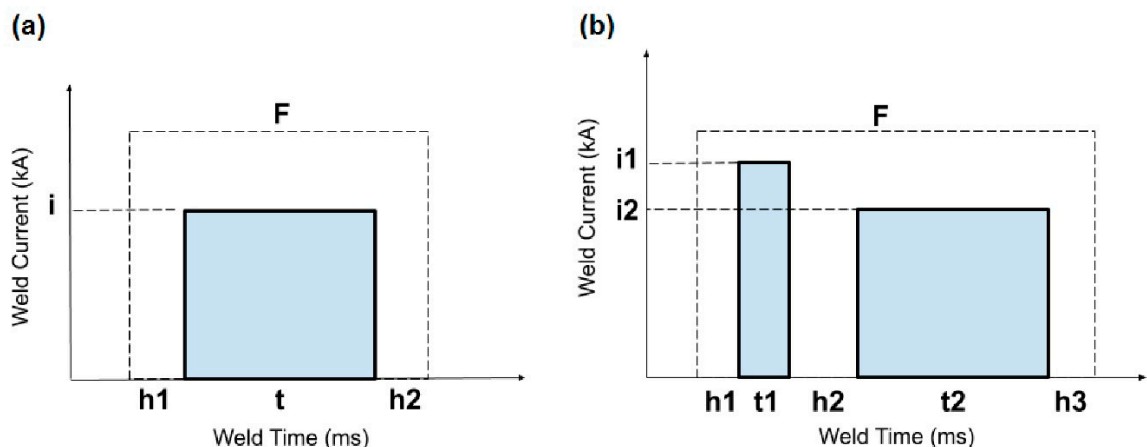

**Figure 2.** RSW schedules: (**a**) One-step; (**b**) two-step.

**Table 2.** Welding process parameters for resistance spot welding (RSW) scheme.

| One-Step Process | | |
|---|---|---|
| F | Weld force, kgf | 400 |
| h1 | Squeeze time, ms | 1000 |
| t | Weld time, ms | 250 |
| h2 | Holding time, ms | 200 |
| i | Weld current, kA | 5.0, 5.5, 6.0 |
| **Two-Step Process** | | |
| F | Weld force, kgf | 400 |
| h1 | Squeeze time, ms | 1000 |
| t1 | Preheat weld time, ms | 60 |
| h2 | Holding time, ms | 15 |
| t2 | Weld time, ms | 250 |
| h3 | Holding time, ms | 200 |
| i1 | Preheat weld current, kA | 6 |
| i2 | Weld current, kA | 5.0, 5.5, 6.0 |

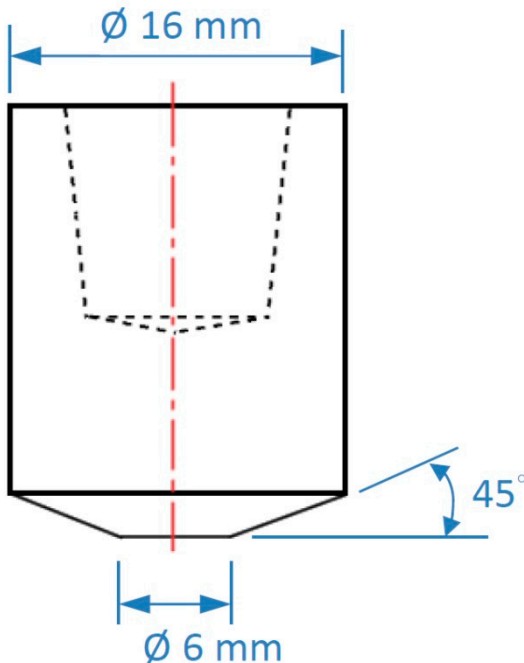

**Figure 3.** The geometry of the customized electrode tip.

The mechanical properties are determined by the tensile shear test, which is most commonly used for the determination of the spot weldment properties in the automotive industry due to its convenience. The test specimens were prepared with dimensions of 105 mm × 45 mm, with an overlap length of 35 mm. The tests were performed using a 100-kN capacity GOTECH AI-7000 material testing system (GOTECH Inc., Taichung) with a crosshead speed of 5 mm/min. The tensile–shear load and fracture energy were determined after the test. The nugget diameter was measured according to AWS D8.1M. The hardness distribution was measured using a Vickers hardness tester (Mitutoyo Inc., Kawasaki, Japan) with a load of 500 g for 10 s, and the hardness values were measured along the weldment at regular intervals of 0.2 mm in the long transverse direction of the original sheets. The samples were ground and polished following standard metallographic procedures. The polished specimens were etched with 2% Nital to reveal the microstructures using an optical microscope LEXT OLS 410 (OLYMPUS Inc., Tokyo, Japan) and a scanning electron microscope Zeiss sigma 300 (ZEISS Inc., Oberkochen, Germany). The failure mode was confirmed after the tensile shear test, and the macrostructures were observed using OLYMPUS DSX 110 (OLYMPUS Inc., Tokyo, Japan).

## 3. Results and Discussion

### 3.1. Nugget Diameter

Figure 4 shows the variations in the 15B22 nugget diameter corresponding to the weld currents from 5.0 to 6.0 kA by the one-step and two-step RSW processes. Both procedures show that the nugget diameter increases with increasing weld current. However, the two-step RSW appears to have a larger diameter than that of the one-step RSW, particularly at higher current.

The heat input is closely related to the weld current. The relationship can be simplified by Joule's law, $H = I^2 Rt$. The formula shows that the heat input significantly increases as current increases. Also, Rao et al. [16] and Lin et al. [17] demonstrated that the growth of the nugget diameter depends on the value of the heat input. Hence, the nugget diameter increases with the weld current. In the two-step RSW, the current is identical to that in the one-step RSW; however, the weld time in the former is higher than that in the latter, thus resulting in greater heat inputs. Therefore, the weld diameters of the two-step samples are larger than those of the one-step samples.

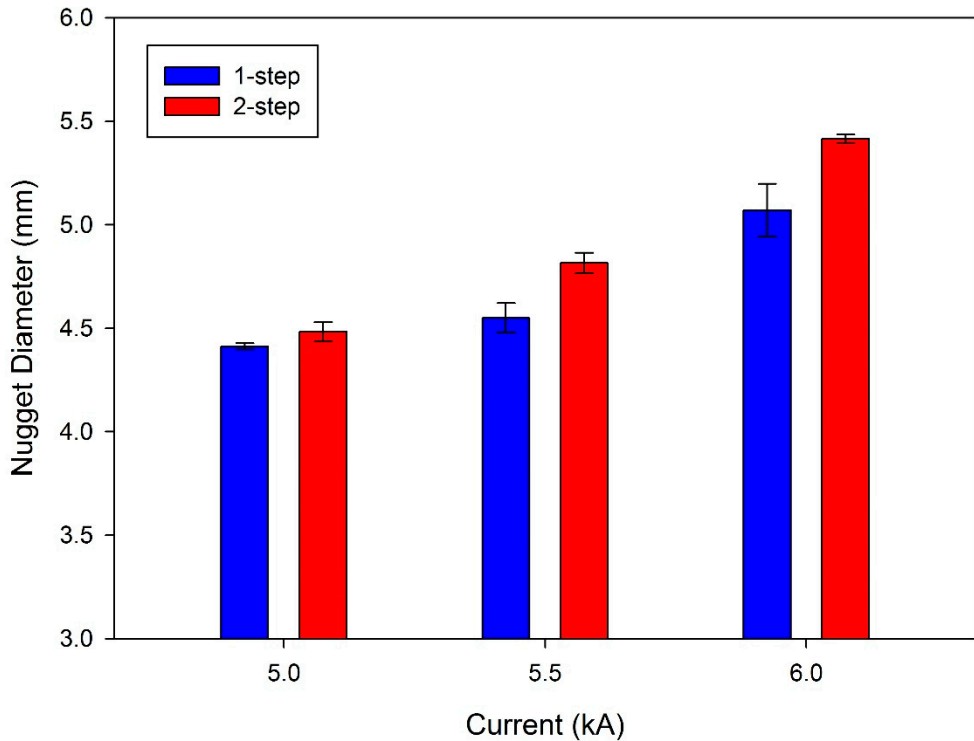

**Figure 4.** 15B22 nugget diameter variations at weld current from 5.0 to 6.0 kA in the one-step and the two-step RSW.

### 3.2. Mechanical Properties

Figure 5a shows the tensile–shear load variation in the 15B22 for different currents. The tensile–shear load of the two-step RSW is higher than that of the one-step RSW with welding at 5.0 kA, but lower at both 5.5 and 6.0 kA. Likewise, the fracture energy of the two-step RSW is higher than that of the one-step RSW by welding at 5.0 kA but lower at 5.5 and 6.0 kA as shown in Figure 5b. An apparent transition from 5.5 to 6.0 kA is clearly indicated by the tensile–shear load and fracture energy measurement.

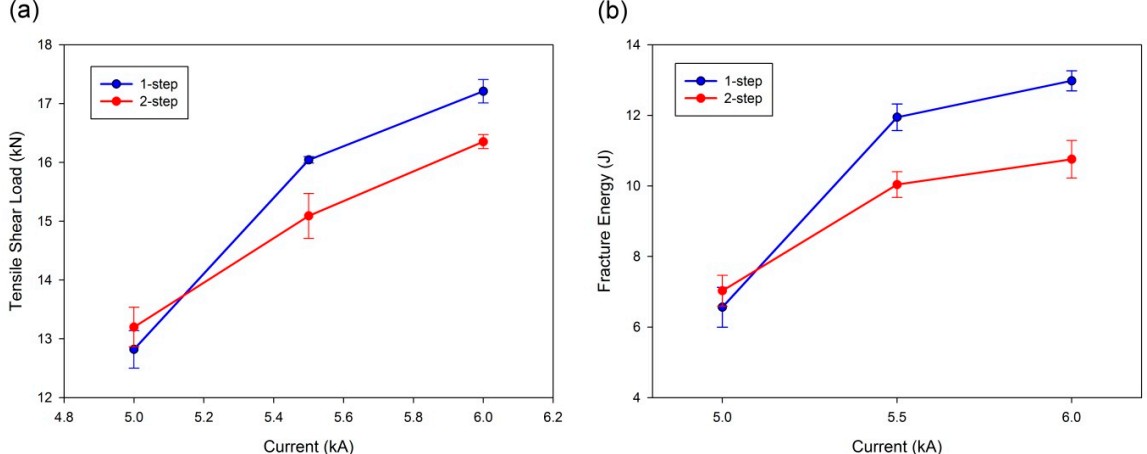

**Figure 5.** Mechanical properties of one-step and two-step RSW specimens: (**a**) Tensile shear load; (**b**) fracture energy.

As mentioned previously, the higher weld current and the weld time both increase the heat input and the nugget diameter, and the increased nugget diameter enhances the loading capability. Liu et

al. [18] got the same result in his study and showed that there is a positive linear correlation between the nugget diameter and the load. However, for 15B22, the tensile–shear load of the two-step RSW at 5.5 and 6.0 kA is lower than that of the one-step RSW. This trend is observed regardless of the fact that the two-step RSW possesses a larger nugget diameter, which indicates an enhanced loading. Accordingly, it could be inferred that there are additional factors that cause the load to decrease, which will be discussed in the following sections.

### 3.3. Hardness Measurement

Figure 6 shows the hardness distribution of the weldments produced by different processes. The trend of hardness distribution was approximately identical for the two different procedures. On average, the base metal has a hardness of HV440 ± 7, whereas the weldment has a higher hardness of ~HV500–HV550. However, a characteristic decline in the hardness can be observed in the curve, as shown in Figure 5, which occurs in the HAZ of the weldment. Figure 7 shows the microstructure of the weldment by the two-step RSW procedure. Apart from the base metal (BM) and fusion zone (FZ), the change in the microstructure in the HAZ is clearly observed in Figure 7. Moreover, according to the degree of the heat influence, the heat affected microstructure can be further classified from the FZ to the BM into the partially melted zone (PMZ), coarse-grain heat affected zone (CGHAZ), fine-grain heat affected zone (FGHAZ), inter-critical heat affected zone (ICHAZ), and the sub-critical heat affected zone (SCHAZ).

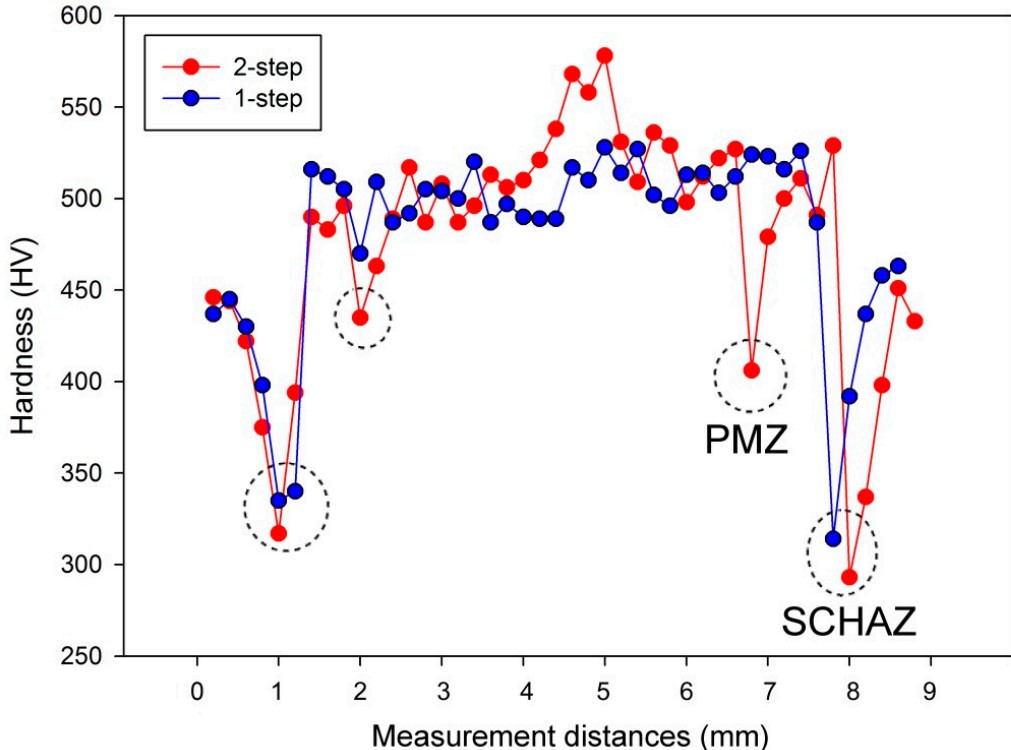

**Figure 6.** Hardness distribution of weldment at weld current of 6.0 kA in the one-step and the two-step RSW.

Figure 8 shows the hardness of PMZ in the HAZ adjacent to the FZ, where the sample is welded with the two-step RSW at 6.0 kA. The diamond-liked indentations show the hardness distribution from FZ over PMZ to HAZ. The PMZ has the relatively low hardness of HV406, compared to those of HV512, 522, 527 in FZ and the HV479, 500 in HAZ. The relatively low hardness causes the comparatively lower strength in this region. The dashed line represents the PMZ, which can be identified both by the hardness measurement and the microstructure variation. Figure 9 shows the hardness of the SCHAZ at

different weld currents. The hardness of both the one-step and the two-step RSW decrease slightly with the increasing weld current. However, the hardness of the two-step RSW declines more than that of the one-step RSW. Figure 10 shows the variation in the local hardness of the SCHAZ at 6.0 kA in greater detail, where the hardness of the BM is ~HV440. Both processes show the occurrence of a dramatic decline in the hardness to below HV300, followed by a gradual increase to HV440. The two-step RSW shows even lower hardness compared with that of the one-step RSW. From the mechanical property measurement, it is clear that two softening regions exist, characterized by the relatively lower hardness in the HAZ of the 15B22 weldment, namely the PMZ in the HAZ adjacent to the FZ and the SCHAZ in the HAZ adjacent to the base metal.

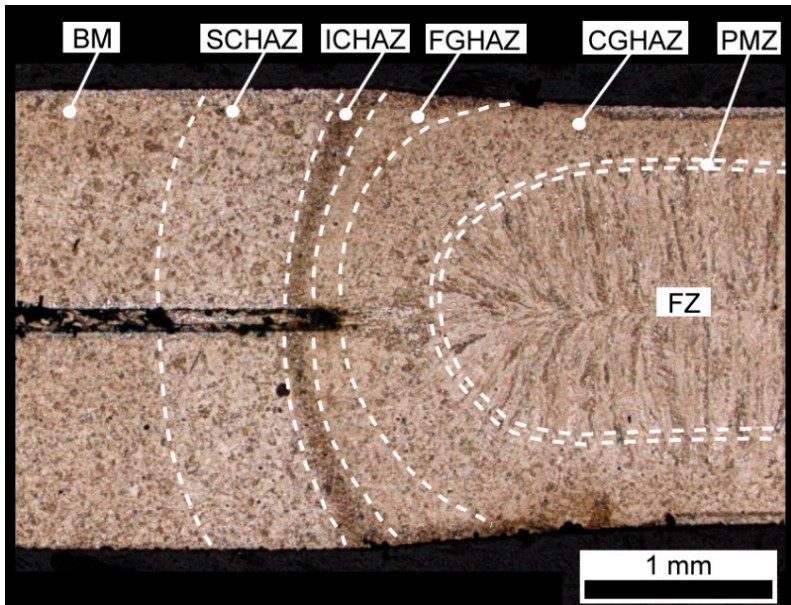

**Figure 7.** Macrostructure region of the specimen in the two-step RSW.

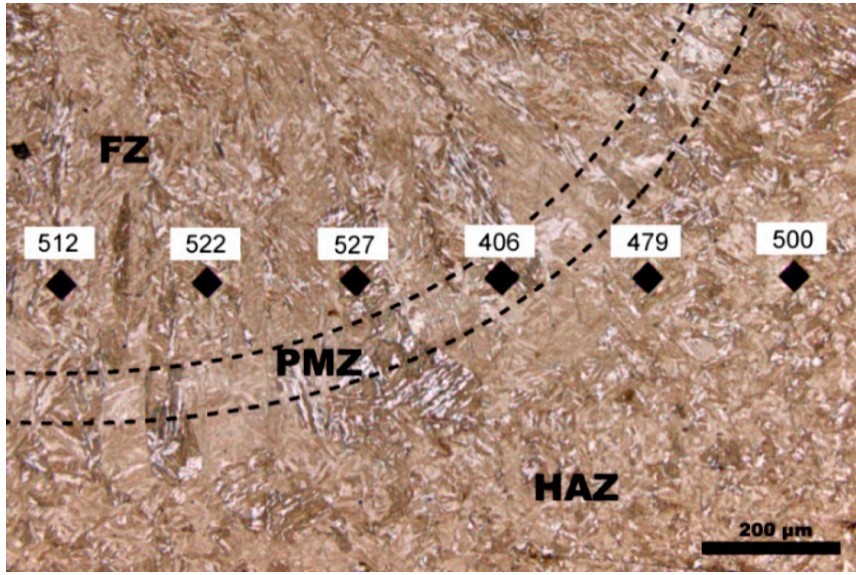

**Figure 8.** Hardness distribution in nearby partially melted zone (PMZ) at 6.0 kA in the two-step RSW.

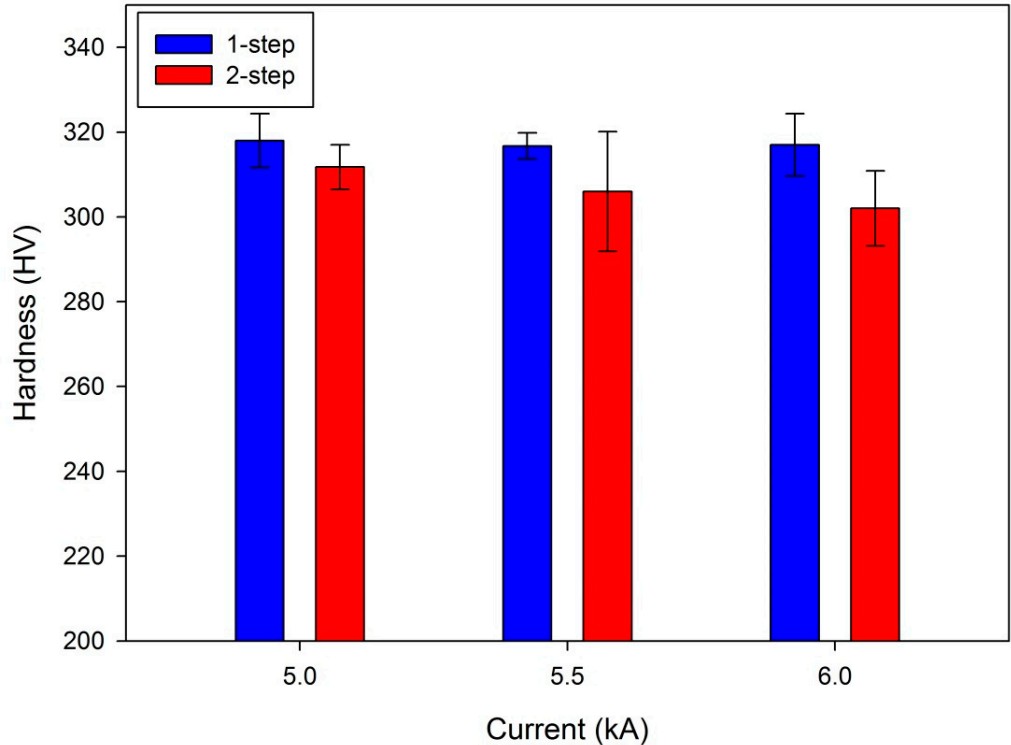

**Figure 9.** Hardness of the sub-critical heat affected zone (SCHAZ) in the one-step and the two-step RSW.

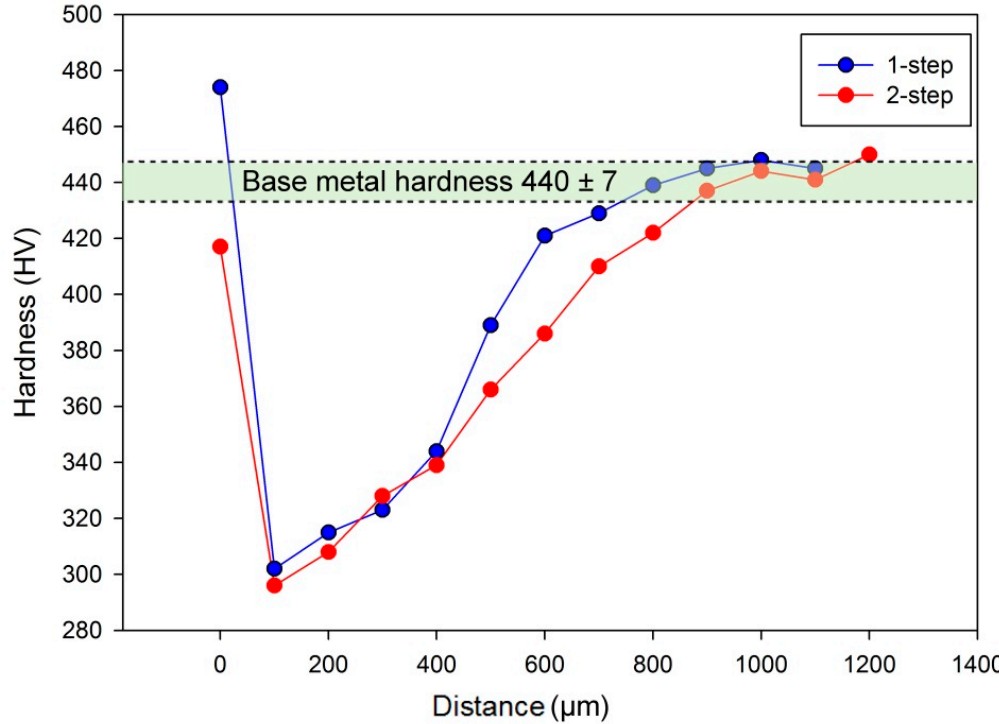

**Figure 10.** Hardness distribution of the SCHAZ at 6.0 kA in the one-step and the two-step RSW.

For the PMZ, the properties and dimensions are greatly influenced by the welding conditions. Huang et al. [19] proposed the relationship between the welding condition and the dimension of the PMZ. They stated that the width of the PMZ increased with the decrease of the average temperature gradient across the width of the PMZ if the temperature difference of the pasty range was a constant. In our study, a wider PMZ during the two-step RSW process could be assumed according the above

because a greater heat input may decrease the average temperature gradient. Conversely, the one-step RSW has a narrower PMZ. Chabok et al. [20] reported that the two-step RSW has two effects on the property of the PMZ. First, the higher heat input would reduce the residual stress of the PMZ and eliminates the compressive stress perpendicular to the plane of the crack. Second, the PMZ of the two-step RSW shows a lower density of high-angle grain boundaries. Thus, it could be inferred that the crack propagation resistance of the PMZ might be lower in the two-step RSW than that in the one-step RSW. Cracking is easily triggered in the SCHAZ due to the lower hardness.

The SCHAZ in the HAZ is another weakness in the 15B22 weldment. Zhou et al. [21] used the finite element method to evaluate the loading behavior and found that the properties and dimensions of the HAZ were predominantly related to the tensile shear load capacity and the failure energy capacity. Zhang et al. [22] mentioned that the strength of the weldment could be enhanced by the improvement of the tensile stress and the yield stress in the HAZ. Particularly, a lower hardness in the SCHAZ is proven to be detrimental to the tensile–shear strength where the cracking is easily triggered.

### 3.4. Microstructure

According to the influence of heat, the microstructure of 15B22 after spot welding can be classified into the BM, HAZ, and FZ. Further according to the thermal history in the HAZ, we can subdivide the HAZ into the CGHAZ, FGHAZ, ICHAZ, and SCHAZ, as shown in Figure 7. With higher magnification, the detailed microstructure of HAZ in optical micrograph is shown in Figure 11. Figure 11a depicts the microstructure of the FZ in the two-step RSW. High temperature above the liquidus line and fast cooling as well causes epitaxial solidification and grain growth towards the weld center and formed the martensite. Previous studies have indicated that shrinkage and solidification cracks are the two main welding defects that may occur in the FZ [23,24]. Figure 11b shows the microstructure of the PMZ in the one-step RSW. The fusion boundary between the FZ and the HAZ can be observed. Figure 11c shows the PMZ microstructure in the two-step RSW. The PMZ is located at the junction of the FZ and the HAZ. It experienced high temperature that caused microstructure to change. This area is clearly characterized by a whitened band seen under the optical microscope, which can be compared to the area with none of the PMZ in the one-step RSW. Figure 11d demonstrates the CGHAZ microstructure in the two-step RSW, which is the region where the temperature greatly exceeds $Ac_3$. Austenization promptly occurs with grain growth. By subsequent cooling the coarsened austenite transforms to martensite with apparent coarse grain. The FGHAZ is the region that experiences the same austenization procedure; however, at a lower temperature above $Ac_3$ compared with that of CGHAZ. Thus, it has a finer austenite grain which was later transformed into a finer martensite, as shown in Figure 11e. The microstructure of the ICHAZ in the two-step RSW shown in Figure 11f is composed of the mixture of bright ferrite and gray martensite. It is the region where the temperature reaches to between $Ac_1$ and $Ac_3$. Depending on the volume fraction of ferrite formed, the hardness of the microstructure is correspondingly lowered. Figure 11g shows the microstructure of the SCHAZ, in which the temperature experienced is below $Ac_1$; instead of a phase transformation there is a tempering effect. Worse is the region where the tempering temperature is adjacent to $Ac_1$, just below the $Ac_1$. Relative high temperature causes severe softening which can be observed by the hardness test. The microstructure of the BM in the two-step RSW where the tempered martensite dominates is shown in Figure 11h. In general, the microstructure under optical microscope revealed no difference between the one-step RSW and the two-step RSW except for the PMZ.

The temperature history of the PMZ passes through the pasty range of the temperature between the liquid phase and the δ ferrite. By equilibrium cooling, the liquid phase and the δ phase would transform into a γ phase via the peritectic reaction. Zhao et al. [25] highlighted that during the rapid cooling of spot welding, the solid–liquid interval would produce segregation due to the redistribution of solute, resulting in a different phase composition. Figure 12a shows the microhardness and its indentation on microstructure of the PMZ in the two-step RSW, and Figure 12b shows an enlarged view. A signification hardness drop is observed in this region. Figure 13 shows an overview of the

PMZ under SEM. The scope of the PMZ, where showed whitened characteristics, was illustrated by dashed lines. The obvious ferrite can be found within the PMZ. Figure 14 shows the enlarged morphology of the PMZ. The ferrite formation adjacent to the prior austenite grain boundary (γ-GB) can be observed. Soomro et al. [26] reported a similar observation after applying double pulse RSW on HSLA. They found that the ferrite can be observed in the outer periphery of FZ during the two-step RSW. Sherepenko et al. [12] explained that the carbon segregation occurred at the fusion boundary, resulting in the softening. During a higher energy input, a diffusional transformation from γ to δ or a nucleation from L to δ is possible according to the phase diagram, leading to the formation of δ ferrite. Sherepenko et al. [13] used the phase field modeling to investigate the behavior of the fusion boundary on 22MnB5 and further confirmed that the segregation caused by the solubility differences between liquid and solid lead to the carbon redistribution in the PMZ.

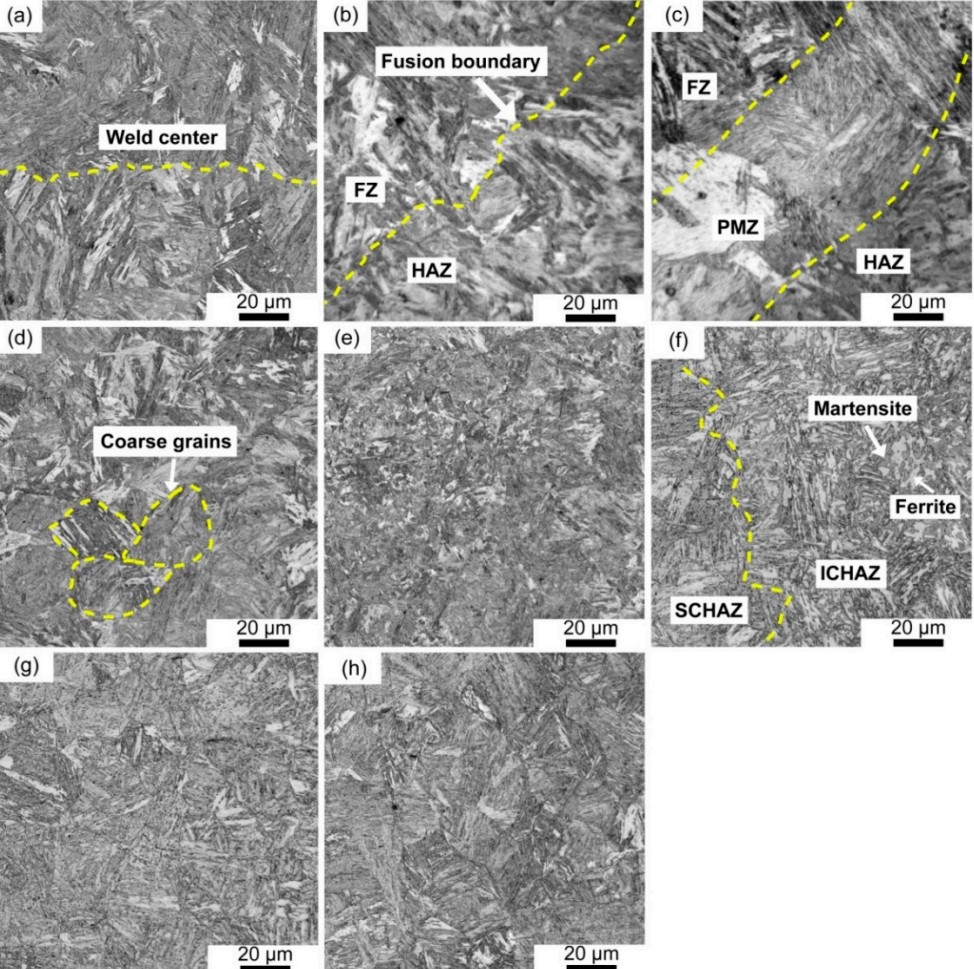

**Figure 11.** Optical photographs showing microstructures of the specimens: (**a**) Fusion zone (FZ) in the two-step RSW; (**b**) PMZ in the one-step RSW; (**c**) PMZ in the two-step RSW; (**d**) coarse-grain heat affected zone (CGHAZ) in the two-step RSW; (**e**) fine-grain heat affected zone (FGHAZ) in the two-step RSW; (**f**) inter-critical heat affected zone (ICHAZ) in the two-step RSW; (**g**) SCHAZ in the two-step RSW; (**h**) base metal (BM) in the two-step RSW.

The tempering of the martensite at high temperature is believed to be the main cause of the SCHAZ softening where the volume fraction of martensite and the degree of tempering impact the softening effect [10,27]. Khan et al. [9] indicated that increasing the volume fraction of the martensite would influence the degree of softening. The hot stamped steel added with boron results in a full martensitic structure. Therefore, a higher degree of softening than the dual phase steel is to be expected.

Pouranvaria et al. [28] concluded that the increase in heat input would also promote the effect of tempering, thus, the two-step RSW has a greater softening effect. Figure 15 shows the metallographic change of the SCHAZ with different degrees of softening. Figure 15a shows the martensite formation in base metal by welding current at 6.0 kA in the one-step RSW. Figure 15b is the SCHAZ with more tempered martensite and more carbon precipitation at 6.0 kA in the one-step RSW. Figure 15c shows the SCHAZ at 6.0 kA in the two-step RSW. A large amount of the tempered martensite, and fine carbides precipitate, the packet boundary, and the block boundary are found in micrograph. Decomposition of martensite at high tempering temperature is apparent in the crystal grains. Several studies reported that the rapid cooling of non-isothermal tempering such as spot welding delayed the formation of spheroidized carbides and the recrystallization process. Conversely, more fine carbides were precipitated [28,29]. Biro et al. [30] also observed that the degree of the martensite decomposition increased as the heat input increased. The process of the two-step RSW obviously increases the heat input and causes a comparatively softer temper martensite to form in the SCHAZ, resulting in a decrease in hardness and tensile–shear strength as well.

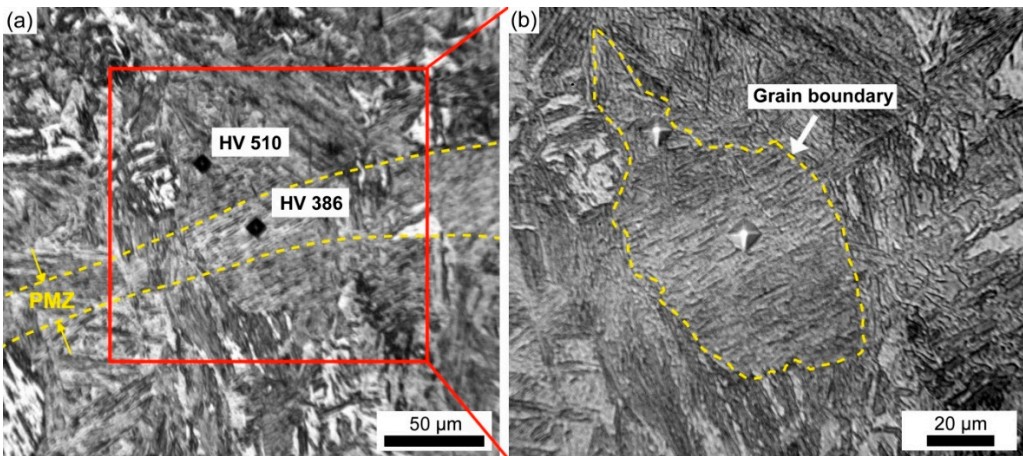

**Figure 12.** The microhardness in the different zones in the two-step RSW under OM: (**a**) Overall view; (**b**) enlarged view.

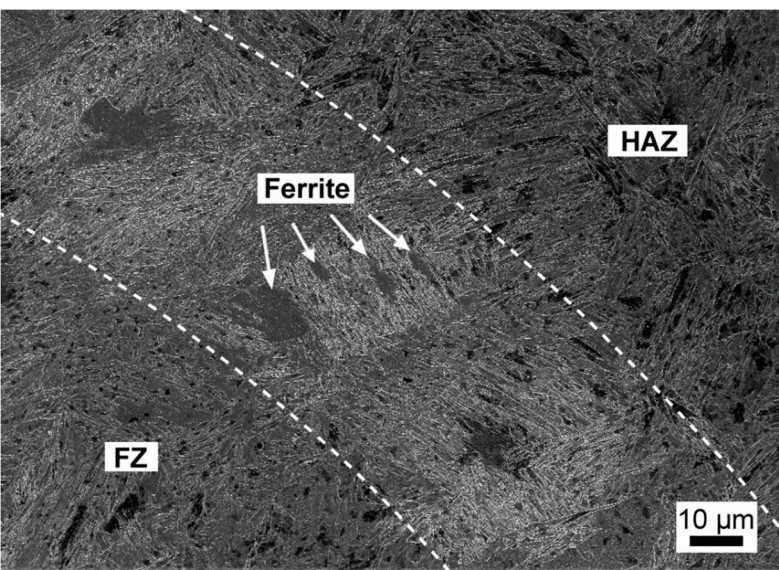

**Figure 13.** An overview of the PMZ under SEM.

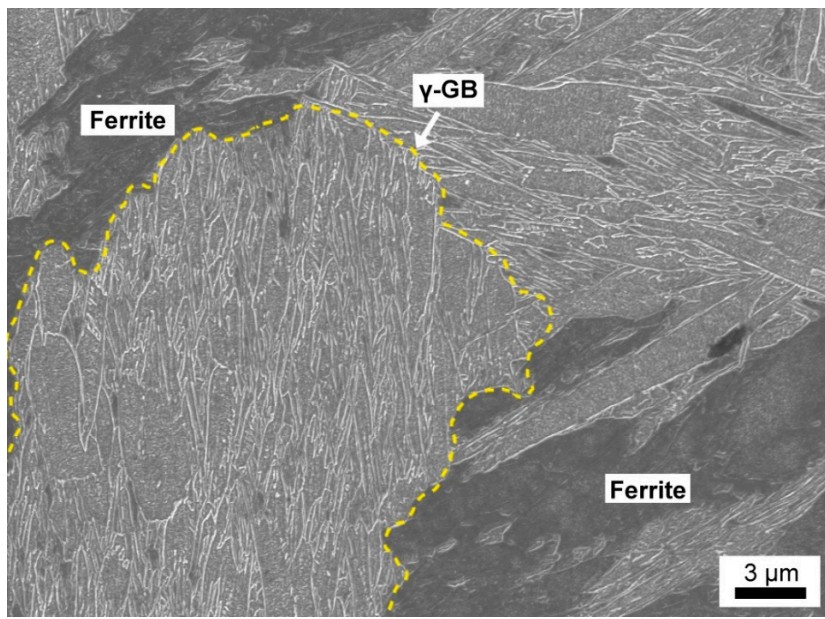

**Figure 14.** The microstructure in the PMZ under SEM.

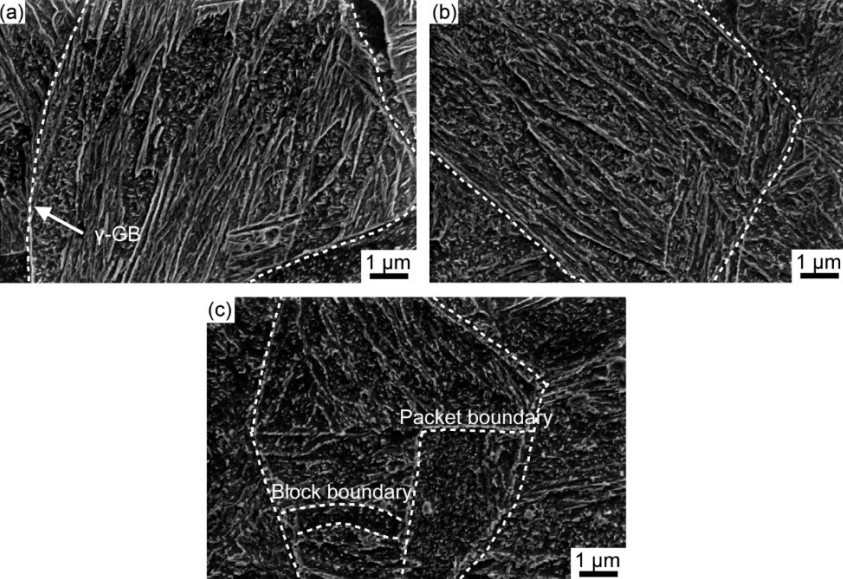

**Figure 15.** The metallographic change of the SCHAZ with different degrees of softening (**a**) BM at 6.0 kA in the one-step RSW; (**b**) SCHAZ at 6.0 kA in the one-step RSW; (**c**) SCHAZ at 6.0 kA in the two-step RSW.

*3.5. Failure Mode*

Depending on the heat input, the 15B22 experiences three different types of failure modes. Figure 16a shows the interfacial failure (IF) mode, where the flat fracture surface is the weld nugget interface. Figure 16b indicates a partial thickness-partial pullout (PT-PP) failure mode and a distinct weld nugget can be observed. Figure 16c shows a pullout failure (PF) mode. The weld nugget remains jointed and the fracture is in its peripheral position. Figure 16d is the PF side view showing the fracture origin and the torsional effect caused by tensile shear. In the observation experiment of the failure mode, three specimens prepared for each welding parameter. All of three specimens showed the same failure mode in different weld conditions. One of them was taken for the representation, as shown in Figure 17. Figure 17a,b show the failure mode at a current of 5.0 kA and both the one-step RSW

and the two-step RSW present in the IF mode. A current of 5.5 kA (refer to Figure 17c,d) exhibits IF and PT-PP fractures using the one-step and two-step RSW processes, respectively. A current of 6.0 kA, (refer to Figure 17e,f) exhibits PT-PP and PF fractures by the one-step and two-step RSW processes, respectively.

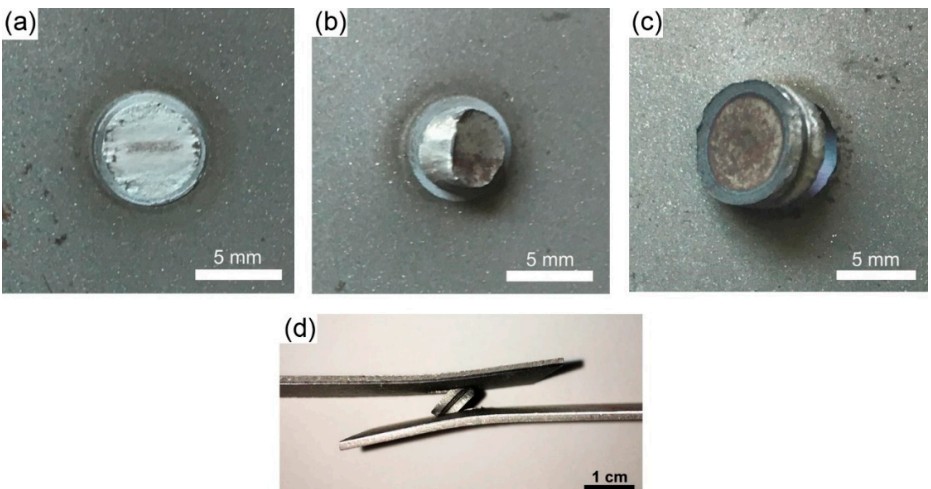

**Figure 16.** Macrostructures of failure mode: (**a**) Interfacial failure (IF); (**b**) partial thickness-partial pullout (PT-PP); (**c**) pullout failure (PF); (**d**) the side view of PF.

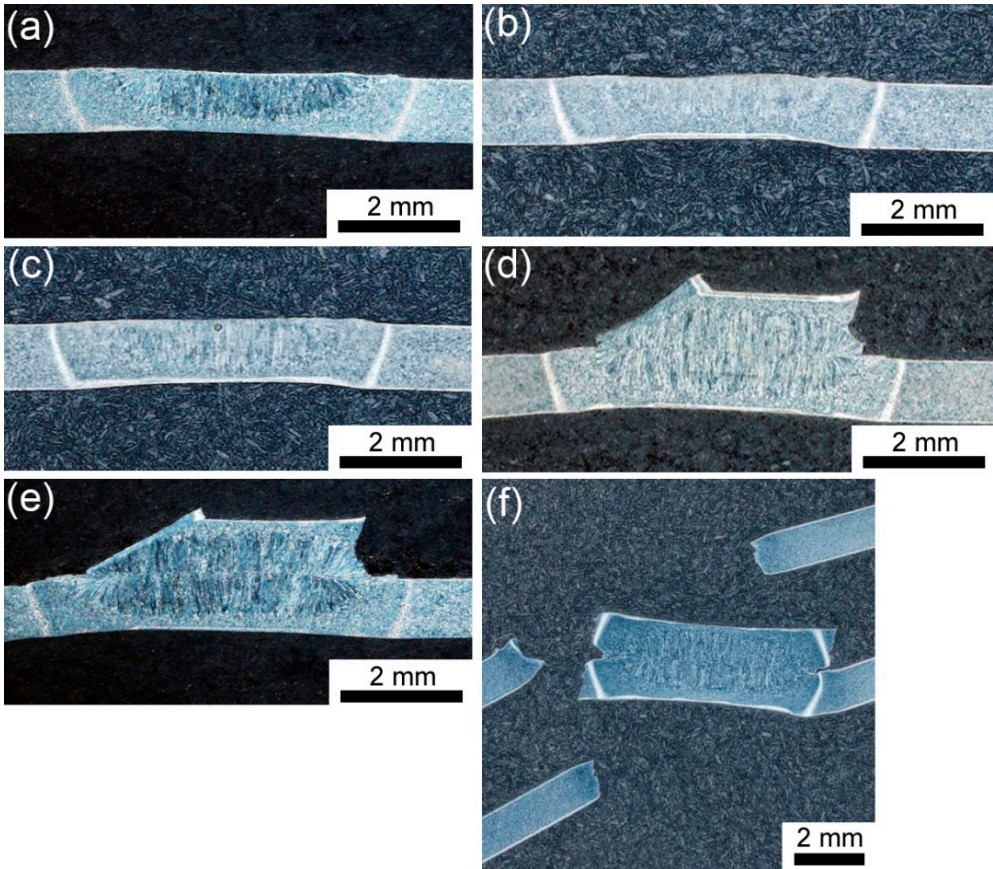

**Figure 17.** Macrostructures showing failure mode at weld current of (**a**) 5.0 kA in the one-step RSW; (**b**) 5.0 kA in the two-step RSW; (**c**) 5.5 kA in the one-step RSW; (**d**) 5.5 kA in the two-step RSW; (**e**) 6.0 kA in the one-step RSW; (**f**) 6.0 kA in the two-step RSW.

During the tensile shear test, the weldment would rotate to make the external force collinear, so the weldment is subjected to a mixture of bending and shear simultaneously [31]. In Figure 16d, the crack initiated at the joint of the two plates adjacent to the weld nugget. Stress concentration caused by the notch effect plays an important role in crack initiation [32]. Figure 18 is the schematic illustration of the failure mode that occurred on the AHSS. Crack initiation and propagation depends on the degrees of heat input. Low heat input makes the weld nugget too small to bear the loading, the crack would initiate at the joint where stress concentration at the tip and causes crack to develop and propagate along the welding center, as shown in Figure 18a. As the heat input increases and the weld nugget becomes larger, the weldment begins to rotate and is mainly subjected to tensile loading. Therefore, tensile stress is dominant and the crack would likely propagate along the PMZ where lower crack resistance appears. Cracking develop and propagate around the weld nugget, as shown in Figure 18b. Mohamadizadeh et al. [11] also speculate that the shear localization at the softening region in the PMZ is triggered by the higher stress concentration at the weld notch. When a much higher heat input is applied, the tensile stress causes the crack to happen in the SCHAZ, and bending moment causes the weldment to rotate, as shown in Figure 18c. Studies have shown that the PF demonstrates a higher plastic deformation and energy absorption than the IF [3]. The two-step RSW results more heat input which changes the size of the weld nugget, the properties of the HAZ, and impacts the failure mode and the mechanical property. In summary, at the weld current of 5.0 kA, the IF failure mode is observed as the weld nugget remains small in size. When the current is increased to 5.5 kA, the failure mode of the two-step RSW transitions into the PT-PP. Similarly, with the high heat input at 6.0 kA, the softening of the SCHAZ becomes severe, and the soft zone becomes the defect in the weldment. Consequently, the two-step RSW with the wider weld nugget has a lower mechanical property than the one-step RSW.

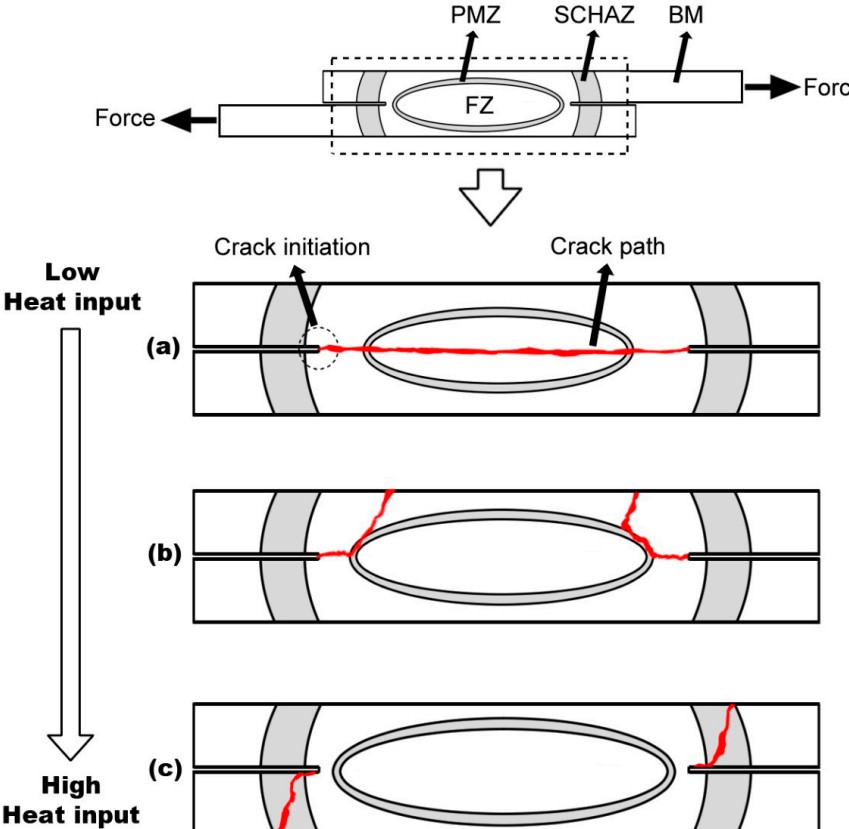

**Figure 18.** Failure schematic illustration of advanced high strength steel (AHSS): (**a**) Lower heat input; (**b**) middle heat input; (**c**) higher heat input.

## 4. Conclusions

Two-step RSW is often applied for joint bonding of AHSS in practical application. This study investigates the effect of two-step RSW on the mechanical properties of 15B22 weldments. The experimental results support the following main conclusions:

1. When the two-step RSW is applied, the additional heat input would increase the size of the weld nugget; however, it would also increase the degree of the softening effect in some certain areas, namely PMZ and SCHAZ. Sudden hardness drop caused by microstructural change results softening effect which lowered the tensile–shear strength in tensile shear test.

2. For 15B22, mechanical properties of the two-step RSW with current of 5.5 and 6.0 kA is worse than that of the one-step RSW. Microstructure investigation reveals that softening of PMZ is due to the formation of larger amounts of soft ferrite in rapid solidification process, and the softening of SCHAZ is due to the tempering martensite formed at high tempering temperature near $Ac_1$. Moreover, the hardness drop in SCHAZ is severer than in PMZ. Compared to an average hardness of HV400 in BM and HV500–550 in FZ of the weldment by 15B22, the hardness in SCHAZ can drop to a low level of HV300. Consequently, it leads to an unexpected earlier fracture due to the insufficient hardness and strength as well.

3. Heat input is the main factor dominating the fracture mode in the two-step RSW process. At the weld current of 5.5 kA, the weldment would fail in the PMZ and show the PT-PP. For higher welding current, increased heat input causes the failure mode to transit from IF to PF, which is attributed to softening of SCHAZ.

4. The implement of a multi-step RSW process for a high strength steel is becoming more common in practical automotive technologies; however, for 15B22, if high current is employed in RSW process, severe softening in SCHAZ caused by the large amount of heat input should be pay attention to since it will lead to the earlier failure in case of the tensile shear loading.

**Author Contributions:** Conceptualization, H.-T.L. and Y.-C.C.; investigation, Y.-C.C.; writing—original draft preparation, Y.-C.C.; supervision, H.-T.L. All authors have read and agreed to the published version of the manuscript.

**Funding:** This research received no external funding.

**Acknowledgments:** The authors would like to thank China Steel Corporation for supporting the materials and the instruments.

**Conflicts of Interest:** The authors declare no conflict of interest.

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
