# Peer review of "Effect of Double Pulse Resistance Spot Welding Process on 15B22 Hot Stamped Boron Steel"

_metals, doi:10.3390/met10101279_

Round 1
Reviewer 1 Report
The authors have presented a solid interesting research paper, discussing an interesting phenomenon in welds of UHSS. However, they have failed to account for previous studies of this phenomenon by a number of authors (see mandatory amendments for more details). As a result, the presented paper lacks scientific novelty and therefore I cannot recommend it to be published in its present state, conducting additional investigations however would improve the paper quality greatly.
Mandatory amendments
Line 40. A number of authors argue the use of IF as a quality criterion for AHSS and UHSS. Please include this in the discussion.
Lines 45-56 please concentrate on multi-pulse welding of presshardening steel, as this material is being studied in the current paper.
Lines 58-62 Please give a sufficient description of hot stamping process (furnace time and temperature, clamping force) and the coating of material. If material was uncoated, please reference under what atmosphere it was hot stamped.
Lines 78-79
Please give a full description of electrode tips according to a current ISO standard (ISO 5821). A sketch of an electrode tip would be of great help.
Lines 105-107
The authors state, that increasing welding time increases heat input and leads to larger welding diameters, but do not account for greater heat loss when welding with longer welding times. Please include this point not the discussion.
Lines 165-180
The authors show an equation to assess the size of the PMZ. Does this reflection predict the size of the PMZ for steel? Otherwise it is not clear, why use it in the current paper. Please show the measurements of the PMZ and their relevance to the calculated values.
Figure 12
Please show further micrographs of the PMZ with lower magnification, so that the position of the ferrite and martensite could be clearly tracked back to the PMZ region. Please use an analytical method (eg. EBSD) to show the presence of ferrite in the PMZ.
3.3 and 3.4
The PMZ, its occurrence and microstructure as well as its influence on mechanical properties of weld on press hardened steel have been discussed in the literature previously:
Sherepenko, O.; Jüttner, S.: Transient softening at the fusion boundary in resistance spot welded ultra-high strengths steel 22MnB5 and its impact on fracture processes. In: Welding in the World 63 (2019) 1, S. 151–59.
Sherepenko, O.; Kazemi, O.; Rosemann, P.; Wilke, M.; Halle, T.; Jüttner, S.: Transient Softening at the Fusion Boundary of Resistance Spot Welds: A Phase Field Simulation and Experimental Investigations for Al–Si-coated 22MnB5.
Mohamadizadeh, A.; Biro, E.; Worswick, M.: Shear band formation at the fusion boundary and failure behaviour of resistance spot welds in ultra-high-strength hot-stamped steel. In: Science and Technology of Welding & Joining 2 (2020) 1, S. 1–8.
Mohamadizadeh, A.; Biro, E.; Worswick, M.; Zhou, N.; Malcolm, S.; Yau, C.; Jiao, Z.; Chan, K.: Spot Weld Strength Modeling and Processing Maps for Hot-Stamping Steels. In: Welding Journal 98 (2019) 8, S. 241–49.
Zhao, Y.; Zhang, Y.; Lai, X.: Analysis of Fracture Modes of Resistance Spot Welded Hot-Stamped Boron Steel. In: Metals 8 (2018) 10, S. 764.
Li, Y. B.; Li, D. L.; David, S. A.; Lim, Y. C.; Feng, Z.: Microstructures of magnetically assisted dual-phase steel resistance spot welds. In: Science and Technology of Welding and Joining 21 (2016) 7, S. 555–63.
Please include their results in the discussion to better represent the novelty of the current paper.
The authors of the current paper refer to the number of pulses as a main factor for the PMZ formation, however in pervious investigations by Sherepenko et al. the occurrence of the PMZ is caused by the carbon diffusion from the solid metal towards the liquid when nugget growth saturates and fusion boundary remains stationary. So, nugget growth is of importance for PMZ formation, not the number of pulses. Adding the information about the nugget growth (e.g. investigated by stop-action test as in Sherepenko, O.; Jüttner, S.: Transient softening at the fusion boundary in resistance spot welded ultra-high strengths steel 22MnB5 and its impact on fracture processes. In: Welding in the World 63 (2019) 1, S. 151–59.) to the paper would greatly help the understanding of PMZ-formation mechanisms.
Author Response
Response to Reviewer 1 Comments
We are deeply grateful to you for detailed comments on our previous manuscript. The revision has been carried out in response to those comments. The point-by-point responses are given in detail as following. Point-by-point responses to the reviewer’s comment using Red. For your convenience, we attach the revised manuscript for reference. Please see the attachment (Revised manuscript for Reviewer 1).
Point 1: Line 40. A number of authors argue the use of IF as a quality criterion for AHSS and UHSS. Please include this in the discussion.
Response 1: Thank you for your suggestions. The use of IF as a quality criterion is indeed an argument. For AHSS, it may be able to meet the tensile shear force requirement of the standard even in the IF failure. However, the appearance of IF failure means the lower heat input and lower mechanical properties, which should be avoided as much as possible. We have added the description into lines 41-44 in page 1.
Point 2: Lines 45-56. Please concentrate on multi-pulse welding of press hardening steel, as this material is being studied in the current paper.
Response 2: Thank you for your comments, and we fully understand your thoughts. It is important to focus on multi-pulse welding of the hot-stamped steel. However, we try to describe the necessity and effect of multi-pulse welding in the process first, as shown in lines 45-50 in page 2. Then a newer study has been added in lines 54-55 in page 2, which mention the softening position of the hot-stamped steel and meet the scope of our paper well. Lastly, the implementation of multi-pulse spot welding for different welding conditions in the hot-stamped steel, which is the innovation of our paper, is introduced in lines 56-60 in page 2.
Reference: Mohamadizadeh, A.; Biro, E.; Worswick, M. Shear band formation at the fusion boundary and failure behaviour of resistance spot welds in ultra-high-strength hot-stamped steel. Science and Technology of Welding and Joining 2020, 25, 556-563, doi:10.1080/13621718.2020.1773057.
Point 3: Lines 58-62. Please give a sufficient description of hot stamping process (furnace time and temperature, clamping force) and the coating of material. If material was uncoated, please reference under what atmosphere it was hot stamped.
Response 3: Thank you for your suggestions. The description has been added in lines 62-65 in page 2.
Point 4: Lines 78-79. Please give a full description of electrode tips according to a current ISO standard (ISO 5821). A sketch of an electrode tip would be of great help.
Response 4: Thank you for the advices. The electrode tip was customized, but it is quite similar to ISO 5821 Type B0. A sketch has been added for more information about the geometry. Please refer to Figure 3 in page 4. Besides, we apologize for the wrong description regarding the geometry. The correct description should be “truncated cone” instead of “dome shaped”. The sentence has been fixed in lines 85-86 in page 3.
Figure 3. The geometry of the customized electrode tip.
Point 5: Lines 105-107. The authors state, that increasing welding time increases heat input and leads to larger welding diameters, but do not account for greater heat loss when welding with longer welding times. Please include this point not the discussion.
Response 5: We totally agree with your opinion because the heat loss indeed happens during the welding process. We thought about that factor before the experiments. Nevertheless, the quantification of the heat input is a complicated experiment that requires careful design due to the difficulty of thermocouple installation and the extremely short measurement time; in addition, our result showed that the nugget diameter increased with the welding times and the welding currents, which met our predictions. Hence, the factor of the heat loss may not be a critical factor in current study. We do consider that the measurement of the heat input should be our further researches, and thank you for the constructive comments.
Point 6: Lines 165-180. The authors show an equation to assess the size of the PMZ. Does this reflection predict the size of the PMZ for steel? Otherwise it is not clear, why use it in the current paper. Please show the measurements of the PMZ and their relevance to the calculated values.
Response 6: Thank you for the opinions. We quoted the equation for trying to explain the effect of the heat input in the PMZ theoretically. For 15B22, the temperature difference of the pasty range is a constant; therefore, the width of the PMZ increases with the decrease of the average temperature gradient across the width of the PMZ. However, we made a qualitative description instead of a quantitative calculation. The equation has been removed for a clearer description. Please refer to lines 178-182 in page 9.
Point 7: Figure 12. Please show further micrographs of the PMZ with lower magnification, so that the position of the ferrite and martensite could be clearly tracked back to the PMZ region. Please use an analytical method (eg. EBSD) to show the presence of ferrite in the PMZ.
Response 7: We appreciate for the helpful comments. A new micrograph with lower magnification has been added in Figure 13 in page 11 for the clearer overview of the PMZ; additionally, we had added more detailed description in lines 229-232 in page 11 as well. However, we may not be able to introduce further analytical method (e.g. EBSD) because of the support issue of the instrument. In current paper, the presence of the ferrite can be identified by the SEM micrographs and the performance of the hardness.
Figure 13. An overview of the PMZ under SEM.
Point 8: 3.3 and 3.4. The PMZ, its occurrence and microstructure as well as its influence on mechanical properties of weld on press hardened steel have been discussed in the literature previously:
Sherepenko, O.; Jüttner, S.: Transient softening at the fusion boundary in resistance spot welded ultra-high strengths steel 22MnB5 and its impact on fracture processes. In: Welding in the World 63 (2019) 1, S. 151–59.
Sherepenko, O.; Kazemi, O.; Rosemann, P.; Wilke, M.; Halle, T.; Jüttner, S.: Transient Softening at the Fusion Boundary of Resistance Spot Welds: A Phase Field Simulation and Experimental Investigations for Al–Si-coated 22MnB5.
Mohamadizadeh, A.; Biro, E.; Worswick, M.: Shear band formation at the fusion boundary and failure behaviour of resistance spot welds in ultra-high-strength hot-stamped steel. In: Science and Technology of Welding & Joining 2 (2020) 1, S. 1–8.
Mohamadizadeh, A.; Biro, E.; Worswick, M.; Zhou, N.; Malcolm, S.; Yau, C.; Jiao, Z.; Chan, K.: Spot Weld Strength Modeling and Processing Maps for Hot-Stamping Steels. In: Welding Journal 98 (2019) 8, S. 241–49.
Zhao, Y.; Zhang, Y.; Lai, X.: Analysis of Fracture Modes of Resistance Spot Welded Hot-Stamped Boron Steel. In: Metals 8 (2018) 10, S. 764.
Li, Y. B.; Li, D. L.; David, S. A.; Lim, Y. C.; Feng, Z.: Microstructures of magnetically assisted dual-phase steel resistance spot welds. In: Science and Technology of Welding and Joining 21 (2016) 7, S. 555–63.
Please include their results in the discussion to better represent the novelty of the current paper.
Response 8: We are grateful for helping to extend the novelty of the current paper. The studies above meet the scope of our discussion well, and we have read them carefully. The paper “Analysis of Fracture Modes of Resistance Spot Welded Hot-Stamped Boron Steel” has been included in our discussion in lines 225-227 in pages 10-11 in the previous manuscript version and the others has been added in the new manuscript version according to needs. Please refer to the lines 229-232 in page 11, 235-240 in page 11, and 299-300 in page 15.
Reference: Sherepenko, O.; Jüttner, S.: Transient softening at the fusion boundary in resistance spot welded ultra-high strengths steel 22MnB5 and its impact on fracture processes. In: Welding in the World 63 (2019) 1, S. 151–59.
Sherepenko, O.; Kazemi, O.; Rosemann, P.; Wilke, M.; Halle, T.; Jüttner, S.: Transient Softening at the Fusion Boundary of Resistance Spot Welds: A Phase Field Simulation and Experimental Investigations for Al–Si-coated 22MnB5.
Mohamadizadeh, A.; Biro, E.; Worswick, M.: Shear band formation at the fusion boundary and failure behaviour of resistance spot welds in ultra-high-strength hot-stamped steel. In: Science and Technology of Welding & Joining 2 (2020) 1, S. 1–8.
Point 9: The authors of the current paper refer to the number of pulses as a main factor for the PMZ formation, however in pervious investigations by Sherepenko et al. the occurrence of the PMZ is caused by the carbon diffusion from the solid metal towards the liquid when nugget growth saturates and fusion boundary remains stationary. So, nugget growth is of importance for PMZ formation, not the number of pulses. Adding the information about the nugget growth (e.g. investigated by stop-action test as in Sherepenko, O.; Jüttner, S.: Transient softening at the fusion boundary in resistance spot welded ultra-high strengths steel 22MnB5 and its impact on fracture processes. In: Welding in the World 63 (2019) 1, S. 151–59.) to the paper would greatly help the understanding of PMZ-formation mechanisms.
Response 9: As responded in point 8, we have improved the statement by introduced the previous study “Transient softening at the fusion boundary in resistance spot welded ultra-high strengths steel 22MnB5 and its impact on fracture processes” in lines 235-238 in page 11. We are so thankful to you for a helpful comment.

Reviewer 2 Report
- The subject of this paper was to investigate the effect of the softening effect of 2-step RSW on the mechanical properties and fracture mode of the weld in resistance spot welding of hot stamped boron steel 15B22. The authors argued that the weld mechanical properties of 2-step RSW are lower than those of 1-step RSW. However, in the experimental design of the 2-step RSW, the authors fixed the preheat condition to one condition, and the welding condition was performed under three conditions. It is necessary to prove whether the authors' claims are valid even in combinations of various preheat conditions and welding conditions.
Author Response
Response: We are deeply grateful to you for detailed comments on our previous manuscript. The aim of the current paper is to investigate the effect of the two-step RSW in fixed preheat welding conditions. After the confirmation of the mechanism induced by multi-pulse welding, the change of the mechanical properties during different welding conditions can be an interesting further study. Thank you for your comments.
Reviewer 3 Report
Its an interesting study about the RSW of 15B22.
I RECOMMEND THE MANUSCRIPT INTO THE SPECIAL ISSUE: WELDING OF ADVANCED HIGH STRENGTH STEELS
Generally:
-the figures are high quality but some metallography need higher resolution and unified font sizes. Also in some cases the comparison between 1-2 pulse should be presented.
-The results are interesting on their own but they should be also compared with literature data in this strength range e.g. tensile strength, fracture energy....etc.
I made my concrete questions and remarks in the manuscript with reviewers comments.
with appropriate corrections it can be a good paper for metals and also for the recommended special issue.

Author Response
Response to Reviewer 3 Comments
We are deeply grateful to you for detailed comments on our previous manuscript. The revision has been carried out in response to those comments. The point-by-point responses are given in detail as following. Point-by-point responses to the reviewer’s comment using Red. For your convenience, we attach the revised manuscript for reference. Please see the attachment (Revised manuscript for Reviewer 3). The number of line may be different with the previous manuscript because of the revision. We have renumbered all lines according your previous comments “peer-review-8335910.v2.pdf”.
Point 1: Line 107. How was it determined?
Response 1: Thank you for the comments. The nugget diameter was measured according to AWS D8.1M. The description has been added in line 98 in page 4.
Point 2: Lines 118. Then I recommend to put an additional diagram heat input vs. nugget size.
Response 2: We totally understand your concern. The heat input is indeed a critical factor. However, the quantification of the heat input is a complicated experiment that requires careful design due to the difficulty of thermocouple installation and the extremely short measurement time. Therefore, the measurement of the heat input may be our future study.
Point 3: Lines 143. Hardness is given correctly 440 HV0.5. What’s the scatter of the base material?
Response 3: Thank you for the advices. The standard error has been added in line 143 in page 6.
Point 4: Page 6. CGHAZ and FGHAZ are in the UpperCHAZ?
Response 4: In our study, HAZ does not distinguish between upper and lower parts.
Point 5: Figure 6. Please use mm instead to see its size! The 2 step has approx. 25% more heat input, the zones must be significantly larger?
Response 5: Thank you for the suggestions. The figure was remade. Please refer to Figure 6 in page 7. The heat input of the two-step RSW indeed increased; however, the nugget diameter does not have the increase as much as the heat input. Please refer to Figure 4 in page 5.
Figure 6. Hardness distribution of weldment at weld current of 6.0 kA in the one-step and the two-step RSW.
Point 6: Figure 7. Use same font sizes on the image. Which specimen? For a comparison the two methods could be presented as a "mirror" image. Indicate the line for the hardness measurement points!
Response 6: Thank you for the comments. We have fixed the font sizes. Please refer to Figure 7 in page 7; additionally, the description of the specimen has been added as well. In this figure, we would like to focus on introducing the different regions. Besides, the PMZ in the one-step RSW was not clear. Therefore, we show the macrograph of the two-step RSW. On the other hand, the description of the hardness measurement point has been added in lines 99-101 in page 4.
Figure 7. Macrostructure region of the specimen in the two-step RSW.
Point 7: Line 167. Only on one side of the nugget according fig 6, what’s the scatter of the measurement points?
Response 7: The statement “Both processes show the occurrence of a dramatic decline in the hardness to below HV300, followed by a gradual increase to HV440.” refers to Figure 9 in page 8, which is another measurement of the hardness.
Point 8: Figure 8. Please use mm instead to see its size!
Response 8: Thank you for the suggestions, but the scale bar would be too big if we use mm for the unit.
Point 9: Figure 10. Please indicate base material hardness scatter!
Response 9: We have remade the Figure 10 in page 9 and add the standard error on it.
Figure 10. Hardness distribution of the SCHAZ at 6.0 kA in the one-step and the two-step RSW.
Point 10: Line 193. Related how and to what properties?
Response 10: The description has been changed. Please refer to the lines 193-194.
Point 11: Figure 11. Which specimen? Comparison between 1-2 step could be interesting. Scale bars are unreadable enlarge them, and unify font sizes. A bit larger magnification would be better. In Figure 11(a), magnification seems too low. In Figure 11(b), microstructure of PMZ is unreadable, please enlarge.
Response 11: We appreciate for the constructive suggestions. In general, the microstructure under optical microscope reveals no difference between the one-step RSW and the two-step RSW except for the PMZ. Therefore, we made new micrographs for comparing the one-step RSW to the two-step RSW in the PMZ. Besides, the detailed description has been added, and the entire Figure 11 in page 10 has been remade for the clearer demonstration. Please refer to the line 203, the lines 206-208 in page 9; the lines 211-212, the line 218, the lines 224-227 in page 10.
Figure 11. Optical photographs showing microstructures of the specimens: (a) FZ in the two-step RSW; (b) PMZ in the one-step RSW; (c) PMZ in the two-step RSW; (d) CGHAZ in the two-step RSW; (e) FGHAZ in the two-step RSW; (f) ICHAZ in the two-step RSW; (g) SCHAZ in the two-step RSW; (h) BM in the two-step RSW.
Point 12: Figure 12. Which specimen? In Figure 12(a), those indentations are even smaller than on fig 7 with smaller magnification, how can that be? Was the load changed? In Figure 12(b), unify font sizes! The proposed PMZ goes through grains which didn’t look like partially melted, how can it be? The PMZ here is smaller than on fig the difference in 11b? Micro hardness difference could be due to the near grain boundary?
Response 12: The process of the specimen has been pointed out. The load of the indentation indeed changed to 25 g for fitting the size of the microstructure. In Figure 12b in page 11, the font size has been fixed. We have added more discussions of the PMZ formation. Please refer to the lines 245-251 in page 11. The size of the PMZ changes with locations. The most regular width the PMZ in our study is approximately 50 μm according to Figure 11c in page 10 and Figure 13 in page 12. The difference in micro hardness exceeds HV100. It may not be the effect of the grain boundary.
Reference: Sherepenko, O.; Jüttner, S.: Transient softening at the fusion boundary in resistance spot welded ultra-high strengths steel 22MnB5 and its impact on fracture processes. In: Welding in the World 63 (2019) 1, S. 151–59.
Sherepenko, O.; Kazemi, O.; Rosemann, P.; Wilke, M.; Halle, T.; Jüttner, S.: Transient Softening at the Fusion Boundary of Resistance Spot Welds: A Phase Field Simulation and Experimental Investigations for Al–Si-coated 22MnB5.
Figure 12. The microhardness in the different zones in the two-step RSW under OM: (a) Overall view; (b) Enlarged view.
Figure 13. An overview of the PMZ under SEM.
Point 13: Figure 14. Unify font sizes.
Response 13: The font size has been unified in Figure 14 in page 12.
Figure 14. The microstructure in the PMZ under SEM.
Point 14: Line 277. Did the specimens fractured in this zone?
Response 14: Yes, they did. Please refer to the discussion “3.5 Failure mode”.
Point 15: Figure 15. Scale bars unreadable, please unify font sizes!
Response 15: The scale bars and the font sizes in Figure 15 in page 13 have been both fixed.
Figure 15. The metallographic change of the SCHAZ with different degrees of softening (a) BM at 6.0 kA in the one-step RSW; (b) SCHAZ at 6.0 kA in the one-step RSW; (c) SCHAZ at 6.0 kA in the two-step RSW.
Point 16: Figure 17. Please use same magnification! And unified font sizes, excess image borders can be cropped everywhere.
Response 16: The magnification has been adjusted. The font size also has been unified. However, considering the overview of the PF, we would not like to adjust the magnification in Figure 17(f). In addition, we greatly apologize for the unified font sizes and have fixed them all. Thank you for your correction.
Figure 17. Macrostructures showing failure mode at weld current of (a) 5.0 kA in the one-step RSW; (b) 5.0 kA in the two-step RSW; (c) 5.5 kA in the one-step RSW; (d) 5.5 kA in the two-step RSW; (e) 6.0 kA in the one-step RSW; (f) 6.0 kA in the two-step RSW.
Point 17: Figure 18. The results are interesting on their own but they should be also compared with literature data in this strength range e.g. tensile strength, fracture energy....etc. e.g. Shear tension strength of resistant spot welded ultra high strength steels.
https://doi.org/10.1016/j.tws.2019.04.051
Response 17: Thank you for the suggestions. It is really an interesting issue to compare with other literature data; however, the multi-pulse spot welding is a quite new process, and there are no large numbers of studies to compare with. In current paper, we try to focus on explaining the effect of the two-step RSW. Comparing with significant amount of literature data should be a meaningful work in future studies.
Point 18: Line 332. T-shear.
Response 18: It has been fixed in line 332 in page 15.
Point 19: Line 348. In case of shear loading! Other types of loads and cyclic loads were not investigated!
Response 19: It has been fixed in line 348 in page 15. Thank you very much.

Reviewer 4 Report
This paper evaluates the resistance spot weldability of boron steel by applying double pulse welding current process. Overall, the contents and results of the experiment are well described, and the logic is well developed without contradictions academically. So I recommend publishing this paper in this journal.
Author Response
Response: We deeply appreciate your detailed comments on our previous manuscript. This is a great encouragement for us. Thank you very much.
Round 2
Reviewer 1 Report
Lines 39-43 please provide the quote
In the introduction the work of by Mohamadizadeh et al. was quoted, but the investigtions by Sherepenko et al. and Li et al. are not reffered to. It should be made clear, that the softening at the fusion boundary has been extensively studied over the past years. The authors have to present the novelty of the conducted inestigations compared to the literature. Please correct.
Was hot stamping conducted without coating? Was a protective gas atmosphere used? If not, did this in any way affect the chemical composition of the material? Was the chemical composition (Table 1) measured before or after hot stamping? How was it measured? It is known, that decarburization of the material in furnace may take place. Please discuss this fact.
Please give the shot peening parameters. (Abrasive type, air pressure, etc.). Does this represent standard industrial practice? If not, discuss the influence of material preparation when transfering the results into industrial conditions.
How was the removal of oxides confirmed? Were any investigations of the material surface conducted?
Lines 189-195
In the literature (e.g. Sherepenko et al.) the width of the PMZ is linked with the time the fusion boundary is not moving, so that the carbon diffusion can take place. Li refers to movement of the molten metall in the weld nugget that accelerates the segregation process. The authors of the paper claim, change in temperature gradient is responsible for the change in PMZ width, but do not support this claim by any data. The investigations by Huang et al. cited here were done for aluminum. Does a prolonged welding time have the same effect with the same magnitude when welding steel with a much lower thermal conductivity, compared to aluminum? Please support your hypothesis with data.
Lines 222-223 Why is there no PMZ in one-step welds? According to your hypothesis, it still must be present, as the temperature gradient is there and thus is the PMZ. Please discuss.
Lines 263-264
It is unclear what the authors mean
Please discuss the nature of the ferrite, found in the PMZ. Give a clear statement, whether the authors support the hypothesis about the delta ferrite formation by Sherepenko et. al.
Fracture mode
Please give a clear description of how many specimens were tested per each condition and how many of them exhibited the fracture mode, presented in figure 17
Conclusions
The authors conclude, that the use of the two impulses causes the softening in the PMZ due to higher heat input. The heat input however has not been calculated in the present paper and in any way has been connected to the formation to the PMZ except of by the assumpton, that it changes the temperature gradient. Its influence on the nugget growth has not been discussed either. Please support the conclusion by solid data.
Author Response
Response to Reviewer 1 Comments
We are deeply grateful to you for detailed comments on our previous manuscript. The revision has been carried out in response to those comments. The point-by-point responses are given in detail as following. Point-by-point responses to the reviewer’s comment using Red. For your convenience, we attach the revised manuscript for reference, please see the attachment (Revised manuscript for Reviewer 1).
Point 1: Lines 39-43. Please provide the quote.
Response 1: Thank you for your suggestion. The study has been cited in line 44 on page 1.
Point 2: In the introduction the work of by Mohamadizadeh et al. was quoted, but the investigations by Sherepenko et al. and Li et al. are not referred to. It should be made clear, that the softening at the fusion boundary has been extensively studied over the past years. The authors have to present the novelty of the conducted investigations compared to the literature. Please correct.
Response 2: Thank you for your comments. We totally understand your concerns. The novelty of the paper is indeed the most important thing. Comparing mechanical properties between one- and two-step RSW is the main innovation and objective of our study. However, the important investigations should be referred to in the introduction. We thus have added the papers by Sherepenko et al. and Li et al. in lines 55-62 on page 2.
References: Sherepenko, O.; Jüttner, S. Transient softening at the fusion boundary in resistance spot welded ultra-high strengths steel 22MnB5 and its impact on fracture processes. Welding in the World 2019, 63, 151-159, doi:https://doi.org/10.1007/s40194-018-0633-3.
Sherepenko, O.; Kazemi, O.; Rosemann, P.; Wilke, M.; Halle, T.; Jüttner, S. Transient softening at the fusion boundary of resistance spot welds: a phase field simulation and experimental investigations for Al–Si-coated 22MnB5. Metals 2020, 10, 10, doi:https://doi.org/10.3390/met10010010.
Li, Y.B.; Li, D.L.; David, S.A.; Lim, Y.C.; Feng, Z. Microstructures of magnetically assisted dual-phase steel resistance spot welds. Science and Technology of Welding and Joining 2016, 21, 555-563, doi:https://doi.org/10.1080/13621718.2016.1141493.
Point 3: Was hot stamping conducted without coating? Was a protective gas atmosphere used? If not, did this in any way affect the chemical composition of the material? Was the chemical composition (Table 1) measured before or after hot stamping? How was it measured? It is known that decarburization of the material in furnaces may take place. Please discuss this fact.
Please give the shot peening parameters. (Abrasive type, air pressure, etc.). Does this represent standard industrial practice? If not, discuss the influence of material preparation when transferring the results into industrial conditions.
How was the removal of oxides confirmed? Were any investigations of the material surface conducted?
Response 3: The hot stamping was indeed conducted without the coating and the gas protection, and the chemical composition was measured before the hot stamping. In Table 1, the composition was provided by China Steel Corporation. Decarburization could happen during hot stamping; however, for the steel which contains low carbon (0.2wt. %C), the effect of decarburization might not be prominent.
Unfortunately, the shot peening parameters may not be provided because China Steel Corporation was entrusted with the process and the parameters are hard to trace back. We truly apologize for that. The aim of the shot peening is to remove the oxides in our study, which is not a standard industrial practice. Nevertheless, we believe that the process is necessary and helps well for the present study to investigate the effect of the two-step RSW.
We totally comprehend your concern. The oxidation reaction might change the composition on the steel surface during the hot stamping process. In fact, this would be one of the reasons that we introduced the pre-experiment to measure the weld current range in Figure 1. If the oxides had not removed well, the expulsion would have occurred early. Accordingly, the elimination of oxide might be confirmed.
Point 4: Lines 189-195. In the literature (e.g. Sherepenko et al.) the width of the PMZ is linked with the time the fusion boundary is not moving, so that the carbon diffusion can take place. Li refers to movement of the molten metal in the weld nugget that accelerates the segregation process. The authors of the paper claim, change in temperature gradient is responsible for the change in PMZ width, but do not support this claim by any data. The investigations by Huang et al. cited here were done for aluminum. Does a prolonged welding time have the same effect with the same magnitude when welding steel with a much lower thermal conductivity, compared to aluminum? Please support your hypothesis with data.
Response 4: Thank you for your professional comments. As a matter of fact, the formation of the softened region is so quite a complicated issue that should be demonstrated in further study. Several researchers such as Sherepenko et al. and Li et al. have done the outstanding works for explaining the mechanism of the fusion boundary. In our study, the impact of the softening zone and the subsequent effect to the mechanical properties is the main aim. The solid data of the mechanical properties, the hardness distribution, and the microstructure indeed showed the influences of the softed zone. The experiment of the weld diameter additionally indicated the increase of the heat input. Therefore, it can be inferred that the two-step RSW increased the degree of the softened effect due to higher heat input. For the precise description, the conclusions have been fixed in lines 339-340 on page 18. The study by Huang et al. could apply in the steel because of the same mechanism.
Point 5: Lines 222-223. Why is there no PMZ in one-step welds? According to your hypothesis, it still must be present, as the temperature gradient is there and thus is the PMZ. Please discuss.
Response 5: According to the equation provided by Huang et al., the estimation of the PMZ width is a qualitative rather than quantitative measurement. Instead of calculating the explicit width, the equation explains the trend of the change.
Point 6: Lines 263-264. It is unclear what the authors mean. Please discuss the nature of the ferrite, found in the PMZ. Give a clear statement, whether the authors support the hypothesis about the delta ferrite formation by Sherepenko et. al.
Response 6: Thank you for the advice. The ferrite found in the PMZ can be confirmed due to the fact of the hardness and the microstructure. We also support the hypothesis about the delta ferrite formation by Sherepenko et. al. However, for confirming the mechanism of PMZ, more research could be conducted to discuss. In fact, that is our ongoing study.
Point 7: Fracture mode. Please give a clear description of how many specimens were tested per each condition and how many of them exhibited the fracture mode, presented in figure 17.
Response 7: Thank you for your suggestion. The information has been added in lines 295-298 on page 13.
Point 8: Conclusions. The authors conclude that the use of the two impulses causes the softening in the PMZ due to higher heat input. The heat input however has not been calculated in the present paper and in any way has been connected to the formation of the PMZ except by the assumption that it changes the temperature gradient. Its influence on the nugget growth has not been discussed either. Please support the conclusion by solid data.
Response 8: Thank you for your comments. This point is similar to Point 4. We combine both points and then respond together. Please refer to Point 4.